# Microstructure-Based Multiscale Modeling of Deformation in MarBN Steel under Uniaxial Tension: Experiments and Finite Element Simulations

**DOI:** 10.3390/ma16145194

**Published:** 2023-07-24

**Authors:** Yida Zhang, Hong Zhang, Tongfei Zou, Meng Liu, Quanyi Wang, Yubing Pei, Yongjie Liu, Qingyuan Wang

**Affiliations:** 1Failure Mechanics and Engineering Disaster Prevention, Key Laboratory of Sichuan Province, College of Architecture and Environment, Sichuan University, Chengdu 610065, China; scuyidazhang@foxmail.com (Y.Z.);; 2Key Laboratory of Deep Underground Science and Engineering, Ministry of Education, Sichuan University, Chengdu 610065, China; 3State Key Laboratory of Long-Life High-Temperature Materials, Dongfang Turbine Co., Ltd., Deyang 618000, China; 4School of Architecture and Civil Engineering, Chengdu University, Chengdu 610106, China

**Keywords:** MarBN steel, uniaxial tension, microstructure, crystal plasticity, finite element simulation

## Abstract

In the current work, a multiscale model was developed coupling a macro-model with the macromechanical physically based yield strength and a crystal plasticity model with micromechanical properties and realistic grain orientation based on the representative volume element. The simulation results show that the effect of microstructure on the macromechanical properties can be considered in the macro constitutive model due to a good consistency between experimental and computed results; whereas solid strengthening, grain boundaries, and dislocation density played a more crucial role than others. Besides coupling simulation and microstructure by EBSD, the microstructure evolution can be well explained by the micromechanical model. Strain is related to the grain orientation, leading to inhomogeneous deformation, forming the various Schmid factor and slip systems. A plastic strain occurs close to the grain boundaries and declines into the grain, resulting in higher kernel average misorientation (KAM) and geometry necessary dislocations (GNDs) in the grain boundaries. The higher the loading, the higher the local strain. Shear bands with around 45 degrees can be formed, resulting in crack initiation and tensile shear failure. This work has developed the guidance of structural integrity assessment and prediction of mechanical properties for the engineering material and components.

## 1. Introduction

Plastic deformation of polycrystalline alloys is heterogeneous, resulting from strain/stress, misorientation, and microstructure. Addressing plastic deformation heterogeneous and microstructure evolution is critical and can help to understand the physical mechanisms during plastic deformation. Recently, plastic behavior and deformation microstructure have been mainly analyzed based on physical experiments, such as tensile/compression tests and microscopic characterization methods, including scanning electron microscopes (SEM), transmission electron microscopes (TEM), and electron backscatter diffraction (EBSD). Zhang et al. [1] investigated the tensile behavior and microstructure evolution of TiAl alloys and nickel-based single crystals [2,3] using TEM. Furthermore, they used EBSD to analyze the tensile failure mechanism of MarBN steel under various strain rates and temperatures [4]. Besides, Benaarbia et al. [5] used EBSD technology to study the low cycle fatigue behavior of MarBN steel at high temperatures. Xu et al. [6] used EBSD technology and hardness hardening diagram analysis method to study the microstructure evolution behavior of 9% Cr MarBN steel weld. Liu et al. [7] used EBSD technology to study the effect of temperature on microstructure evolution and deformation behavior of in situ EBSD of Fe–32Ni. Although a few experiments have been developed and addressed, there are several drawbacks to the experiments as follows: (1) How to express the deformation history? It is very crucial for strain and stress evolution. (2) How to understand the effect of microstructure on the deformation behavior? (3) It is difficult to define and address the region of interest during deformation history.

Recently, the finite element method (FEM) [8] has been employed to solve the above problems. Furthermore, multiscale constitutive models including the macro- and micromechanical models have been developed to investigate the physical–mechanical responses of materials. Barrett et al. [9] developed a physically based high-temperature yield strength model of 9Cr steels and investigated the effect of various strength mechanisms on yield strength. Dong et al. [10] studied the heterogeneous elastic–plastic properties of 2205 duplex stainless steel welded joints based on the microstructure-based multiscale model. This model is developed by combining a physical mechanism and a crystal plastic model. Engel et al. [9] investigated the effect of grain orientation on the local and global elastic–plastic behavior of nickel-based superalloy. Additionally, Kan et al. [11] developed a multiscale constitutive model for superelastic NiTi shape memory alloys, verified its ability to predict the degradation of tension–compression asymmetry caused by texture, and discussed the effect of texture type on cyclic deformation. Tang et al. [12] developed a set of elastoplastic hysteretic properties considering fatigue damage and used this model to study the hysteretic properties of steel members. Lindroos et al. [13] developed a crystal plastic model, including the phase plastic mechanism of mother austenite and daughter martensite dislocation slip, which can be used to study and calculate the microstructure deformation of polycrystalline aggregates. Yalçinkaya et al. [14] proposed a finite element framework based on crystal plasticity to investigate the mechanical response of the structure and the microstructure evolution of duplex phase (DP) steel under uniaxial tensile loads. However, until now, an investigation into the deformation behavior of MarBN steel based on the multiscale model has been poorly reported.

In the current paper, the tensile experiments for MarBN steel are conducted with the constant strain rate of 5 × 10^−3^ s^−1^ at room temperature (RT). The microstructure of MarBN steel before and after tensile tests was analyzed by EBSD. Meanwhile, a multiscale model was constituted by combining a macro-model with the macromechanical physically based yield strength and a crystal plasticity model with micromechanical properties and realistic grain orientation based on the representative volume element (RVE).

## 2. Experimental Procedures

### 2.1. Materials

The material employed in this manuscript is the MarBN steel (named 9% Cr turbine steel) [15,16,17] with the chemical compositions as follows (Table 1):

To get better mechanical properties, the steel was subjected to the following heat treatment: aging treatment at 1080 °C for 2 h, water cooling, 680 °C for 4 h, and air cooling. An EBSD scan of step size 0.20 μm was used across 1000 μm by 1000 μm field of view to address the microstructure and grain orientations, as shown in Figure 1. From Figure 1a, prior austenite grains boundaries (PAGBs), parallel martensite laths, and coincidence site lattice (CSL) model can be observed. The grain size and orientations vs. relative frequency are plotted in Figure 1b,c where the grain misorientation angle is mainly distributed along 0–10 and 55–65 degrees, including 12% twin boundaries, and the average grain size is 13 μm calculated from EBSD data. Furthermore, the weak texture is addressed in the steel, as shown in Figure 1d.

### 2.2. Uniaxial Tension and Microstructure

The uniaxial tension tests were performed on as-received samples to quantify plastic deformation according to the tension testing international standard [18,19]. Dog bone-shaped samples [15] with a gauge length of 41 mm and diameter of 5 mm were designed, as shown in Figure 2. Uniaxial tension tests are conducted under the constant strain rate at RT, such as 5 × 10^−3^ s^−1^ based on the standard tensile experimental device, as shown in Figure 3. To reduce the data scattering, at least three samples can be used to perform the uniaxial tension at each strain rate. After tension failure, the fracture surfaces at each strain rate can be observed to address the fracture microstructure characterization by the scan electron microscopy (SEM) and EBSD with the scan step of 0.20 μm to analyze the grain orientation, phases distribution, and plastic microstructure behavior.

## 3. Multiscale Constitutive Modeling

### 3.1. Microstructure-Based Macro Constitutive Formulation

#### 3.1.1. Constitutive Formulation Development

On a macroscale, the total strain (ε) includes the elastic ($\varepsilon_{e}$) and plastic strain ($\varepsilon_{p}$), as follows:(1)$$\varepsilon =\varepsilon_{e}+\varepsilon_{p}$$

The stress tensor (σ) and elastic strain ($\varepsilon_{e}$) can be expressed by Hooke’s law as follows:(2)$$\sigma =D_{e}(\varepsilon -\varepsilon_{p})=D_{e}\varepsilon_{e}$$
where $D_{e}$ is the fourth-order elastic tensor related to the elastic modulus (E) and Lamé’s elastic constants/shear modulus (μ) with a value of 221.78 GPa and 85.31 GPa, respectively, [20] as below:(3)$$D_{e}=\frac{E(1-\mu )}{\left(1+\mu )(1-2\mu \right)}\left[\begin{array}{cccccc} 1 & & & & & \\ \frac{\mu }{1-\mu } & 1 & & & & \\ \frac{\mu }{1-\mu } & \frac{\mu }{1-\mu } & 1 & & & \\ 0 & 0 & 0 & \frac{1-2\mu }{2(1-\mu )} & & \\ 0 & 0 & 0 & 0 & \frac{1-2\mu }{2(1-\mu )} & \\ 0 & 0 & 0 & 0 & 0 & \frac{1-2\mu }{2(1-\mu )} \end{array}\right]$$
and the plastic strain can be expressed as follows:(4)$$\varepsilon_{p}={\bar{\varepsilon }}_{p}\frac{\bar{\sigma }}{\sigma }$$
where ${\bar{\varepsilon }}_{p}$ and $\bar{\sigma }$ are equivalent plastic strain increment and equivalent stress, respectively. In addition, in the current work, Von Mises yield rule can be employed to estimate the yield evolution during uniaxial tension as follows:(5)$$F=\sqrt{3J_{2}}-\sigma_{s}=0$$
where $J_{2}$ is second invariant of deviatoric stress tensor and $\sigma_{s}$ is the yield stress. The material parameters used in Section 3.1.1 are all from the literature, and a summary table of these parameters is added at the end of this section (Table 2 and Table 3).

It should be noted that the yield stress is related to the microstructure response based on the internal state variable approach [9,21,22,23]. Therefore, in the present work, the yield stress can be defined as [24,25]:(6)$$\sigma_{s}=\sigma_{ss}+\sigma_{pn}+\sigma_{bd}+\sigma_{ph}+\sigma_{in}+\sigma_{dd}$$
where $\sigma_{ss}$ and $\sigma_{pn}$ are the contribution of solid solution strengthening and Peierls–Nabarro stress, respectively, $\sigma_{bd}$ and $\sigma_{ph}$ are the stress caused by the grain boundaries and precipitates, respectively, $\sigma_{in}$ is strengthening of interstitial atoms, and $\sigma_{dd}$ is the hardening process contributed by the evolution of dislocation density.

Solute atoms can produce a local stress field in the matrix, resulting in an increase in the shear stress of dislocation movement in the vicinity of solute atoms. Therefore, the contribution of n multicomponent substitutional solutes to the whole yield strength can be expressed as [26,27]:(7)$$\sigma_{ss}=\sum_{i=1}^{n}\left[\frac{A}{b}{\left(\frac{F_{m,i}^{4}\varpi }{\Gamma }\right)}^{1/3}{\left(\frac{c_{i}}{l_{0,i}^{2}}\right)}^{2/3}\right]$$
where A and b are a constant with a value of approximately unity with a value of 0.8 [23] and magnitude of Burgers vector of 0.286 nm [28], respectively, $F_{m,i}^{}$ is maximum solute atom–dislocation interaction force, ϖ is the range of interaction, such as 2–3 lattice spacings until yielding occurs [29] in the current work, Γ $c_{i}$ and $l_{0,i}$ are dislocation line tension, identified as $\frac{\mu b^{2}}{2}$ [30], solute concentration, the initial values are defined as 10 for Mn, 0.1 for Si, and 2 for the rest of the alloying elements [31], and mean spacing of solute i, respectively. In addition, the mean spacing of solute ($l_{0,i}$) and maximum solute atom–dislocation interaction force ($F_{m,i}$) can be obtained as:(8)$$l_{0,i}=\left(\frac{2}{c_{i}^{2/3}}-3\right)b_{matrix}+2\left(\frac{b_{matrix}+b_{solute,i}}{2}\right)$$
(9)$$F_{m,i}=\frac{\mu b^{2}}{\varphi }{\left[{\left(\frac{\varepsilon_{\mu }}{1+0.5\left|\varepsilon_{\mu }\right|}\right)}^{2}+{\left(\alpha \varepsilon_{b}\right)}^{2}\right]}^{2/3}$$
where $b_{matrix}$ and $b_{solute,i}$ are magnitude of Burgers vector for matrix and solute i, respectively, φ is a numerical constant, defined as 10 and 11.3 for BCC and FCC materials [9], respectively, $\varepsilon_{\mu }$ and $\varepsilon_{b}$ are the lattice ($\frac{\partial b}{b\partial c}$) and shear ($\frac{\partial \mu }{\mu \partial c}$) modulus misfit parameters, respectively, and α is a constant of material, such as 3–15.

**Table 2 materials-16-05194-t002:** Element parameters for solid solution strengthening [32].

Element	b	μ	ν
α-Fe	0.2482	81.6	0.29
γ-Fe	0.2503	81.6	0.29
Cr	0.2520	115.3	0.21
Al	0.2863	26.3	0.35
Co	0.2507	82	0.31
Mn	0.2667	79.5	0.24
Mo	0.2725	125.6	0.31
Ni	0.2492	76	0.31
Nb	0.2858	37.5	0.4
Si	0.3830	39.7	0.42
Ti	0.2951	45.6	0.32
V	0.2624	46.7	0.37
W	0.2741	160.6	0.28

Peierls–Nabarro stress can lead to move a dislocation in the absence of other strengthening mechanisms, which is expressed based on the Nabarro’s model [33] as follows:(10)$$\sigma_{pn}=\frac{2\mu }{1-v}\exp\left(\frac{-2\pi }{1-\nu }\right)$$
where ν is Poisson’s ratio.

Yield stress caused by grain boundaries includes two parts contribution, such as high angle grain boundaries (HAGBs) [34,35] and low angle boundaries (LABs) [36,37] dislocation substructure, as follows:(11)$$\sigma_{bd}=\sigma_{HAGBs}+\sigma_{LABs}=k_{hall-petch,0}\exp\left(\frac{-T}{T^{*}}\right)\frac{1}{d_{g}^{n_{HAGBs}}}+\frac{\alpha_{2}\mu b}{\frac{2}{\pi }\left[w\ln\left(\tan\left(\arccos\left(w/l\right)+l/w\right)\right)+l\pi /2-l\arccos\left(w/l\right)\right]}$$
where $\sigma_{HAGBs}$ and $\sigma_{\mathrm{LABs}}$ are the contributions of HAGBs and LABs, respectively, $k_{hall-petch,0}$ is the Hall–Petch constant at 0 K with a value of approximately 18.97 $\mathrm{Mpa}\cdot {\mathrm{mm}}^{1/2}$ [38,39,40], $T^{*}$ is a reference constant temperature of room temperature, $d_{g}$ and $n_{HAGBs}$ are the average width of HAGBs (martensitic lath for the block width) with a value of around 500 nm [15,41] and Hall–Petch exponent defined as 0.5, $\alpha_{2}$ is a material parameter ranging from 2 to 3, l and w are the $\frac{1}{3}$ packet with a value about 300 μm and LAB width with a value of approximately 325 nm [17].

Yield strength by precipitates can be defined based on the Ashby–Orowan model [42],
(12)$$\sigma_{ph}=0.045\frac{\mu b}{\lambda }\ln\frac{r}{b}$$
where λ and r are mean interparticle spacing and average radius of precipitates with a value of approximately 1.75 nm [23].

The stress caused by interstitial atoms, such as C and N atoms, is addressed using the simplified assumption that the increase in strength is proportional to the composition of solid solution atoms to the power of 1/2 [9,10,43,44], as follows:(13)$$\sigma_{in}=\frac{1}{M}\sum_{i}\left[k_{c}{\left[C\left(wt.\%\right)\right]}^{0.5}+k_{N}{\left[N\left(wt.\%\right)\right]}^{0.5}\right]$$
where $k_{C}$ and $k_{N}$ are the strengthening coefficient of the various solution atoms.

The yield strength of the hardening process contributed by the evolution of dislocation density can be shown based on the Taylor hardening model,
(14)$$\sigma_{dd}=\alpha_{1}\mu b\sqrt{\rho }$$
where $\alpha_{1}$ and ρ are the material constants between 0.2 to 0.5 and dislocation density ($2.65\times 10^{14}$ m^−2^) [45].

**Table 3 materials-16-05194-t003:** The main parameters used in simulation.

Parameter	Value	Reference
E	282.78 Gpa	[20]
μ	87.5 GPa	
A	0.8	[23] [28]
b	0.286 nm	
$\varphi_{BCC}$	10	[9]
$\varphi_{FCC}$	11.3	
$k_{hall-petch,0}$	18.97 MPa·mm^0.5^	[38,39,40]
$d_{n}$	500 nm	[15]
$n_{HAGBs}$	500 nm	[41]
$\alpha_{1}$	2–3	[17]
λ	1.75 nm	[23]
r	1.75 nm	
$\alpha_{1}$	0.2–0.5	[45]
ρ	2.65 × 1014 m^−2^	

#### 3.1.2. Finite Element Model

A three-dimensional finite element model with the actual tensile specimen was created, in which the full clamping constraint and a displacement load were applied at the bottom and top of the sample in the X-direction, as shown in Figure 4. In addition, to analyze the macro-tensile elastic–plastic behavior considering the microstructure, the microstructure-based macro constitutive formulation was constituted in Abaqus software [46] based on the user-defined material subroutine (UMAT) [47]. The mesh sensitivity was conducted and optimum, and then 64,260 elements and 68,907 nodes were meshed with the element type of C3D8R, as shown in Figure 4.

### 3.2. Crystal Plasticity-Based Constitutive Formulation

#### 3.2.1. Constitutive Law and Finite Element Model

In the current work, plastic deformation is computed by the finite strain theory. Generally, the deformation gradient can be decomposed into the elastic ($F_{e}$) and plastic ($F_{p}$) part [48] as follows,
(15)$$F=F_{e}F_{p}$$

Meanwhile, the velocity gradient ($\tilde{L}$) is expressed as follows,
(16)$$\tilde{L}=\dot{F}F^{-1}={\dot{F}}_{e}F_{e}^{-1}+F_{e}{\dot{F}}_{p}F_{p}F_{e}^{-1}={\dot{L}}_{e}+{\dot{L}}_{p}$$
where $\dot{F}$ is rate of deformation gradient; ${\dot{F}}_{e}$ and ${\dot{F}}_{p}$ are the rate of deformation gradient for elastic and plastic part, respectively; ${\dot{L}}_{e}$ and ${\dot{L}}_{p}$ are the rate of velocity gradient for elastic and plastic part, respectively.

The velocity gradient for plastic part ($L_{p}$) is addressed by the slipping rate ($\dot{\gamma }$) along the activate slip system (α) as follows:(17)$$L_{p}={\dot{F}}_{p}{\dot{F}}_{p}^{-1}=\sum_{\alpha =1}^{N_{slip}}{\dot{\gamma }}^{\alpha }m^{\alpha }⊗n^{\alpha }$$
where $m^{\alpha }$ and $n^{\alpha }$ are the slip direction and normal to slip plane along the activated slip system, respectively. According to the Schmid rule for FCC material, the slipping rate along the activate slip system in a rate-dependent crystalline solid is the function of the critical shear stress [49] as follows:(18)$${\dot{\gamma }}^{\alpha }={\dot{\gamma }}_{0}\left|\frac{\tau^{\alpha }}{\tau_{c}^{\alpha }}\right|\mathrm{sgn}\left(\tau^{\alpha }\right)$$
where ${\dot{\gamma }}_{0}$ is the initial slipping rate; $\tau^{\alpha }$ and $\tau_{c}^{\alpha }$ are the resolved and critical shear stress, respectively; n is the sensitive parameter of slipping.

When the multi-slipping systems are activated, the strain hardening along the activated slip system is addressed by the evolution of the slipping rate (${\dot{\gamma }}^{\beta }$) through the incremental relation as follows:(19)$${\dot{g}}^{\alpha }=\sum_{\beta }h_{\alpha \beta }\left|{\dot{\gamma }}^{\beta }\right|$$
where $h_{\alpha \beta }$ is the slipping hardening moduli over all activated slip systems. $h_{\alpha \beta }$ (α≠β) and $h_{\alpha \alpha }$ (α=β) are defined as latent and self-hardening moduli, respectively, in which the self-hardening moduli can be expressed [49,50] as follows:(20)$$h_{\alpha \alpha }=h_{0}{\mathrm{sech}}^{2}\left|\frac{h_{0}\gamma }{\tau_{s}-\tau_{0}}\right|$$
where $h_{0}$ is the initial hardening modulus; $\tau_{0}$ is the yield stress; $\tau_{s}$ is the stress of large plastic flow initiates; γ is the Taylor cumulative shear strain on all slip systems. In contrast, the latent hardening moduli are expressed by
(21)$$h_{\alpha \beta }=qh\left(\gamma \right)\left(\alpha \neq \beta \right)$$
where q is the material constant, such as 1 and 1.2 [51] for self- and latent hardening moduli, respectively.

In the current work, the FEPX [52] framework was used for the crystal plastic finite element (CPFE) simulation. To explore the micromechanical behavior, the representative volume element (RVE) model is used from the sample working section marked by the red rectangle, as shown in Figure 4a,b. Moreover, to get accurate situational results, the grain orientation and size for MarBN steel presented in Figure 1 were used and integrated into the finite element model. The cube model with a 1×1 mm^2^ cross-section and 2 mm in length was constituted with 100 grains using NEPER [53], as shown in Figure 5b.

#### 3.2.2. Material Properties and Boundary Condition for Micromechanical

According to the results of EBSD presented in Figure 1, the microstructure of MarBN steel includes the primary martensite phase and a small amount of ferrite phase (≤1%). Therefore, the elastic constants are provided for the dual-phase polycrystal, such as martensite (M) and ferrite (α) phases for BCC [54,55,56,57], as shown in Table 4.

Besides, according to the previous reports [58,59,60,61], the initial slip system strengths and other plasticity parameters provided in Equations (1)–(7) can be addressed in Table 5.

To simulate the uniaxial tension behavior, a strain load was applied on the top surface for the Z-axis loading direction presented in Figure 5; in contrast, the other faces were clamped. The maximum strain was set to 10%, and the strain rate was defined as $5\times 10^{-3}$ s^−1^. The boundary conditions are shown in Figure 6.

## 4. Results and Discussion

### 4.1. Mesh Size Sensitive and Model Calibration

To address the effect of element size on microplastic behavior, the RVE model is meshed with different element sizes by the gmsh tool [62] integrated with NEPER using tetrahedral elements of C3D10, as shown in Figure 7. Whereas the element size for RVE model is controlled by the relative cell length value (rcl), such as 0.3 rcl for 472,147 nodes and 341,060 elements; 0.4 rcl for 197,660 nodes and 140,712 elements; 0.6 rcl for 57,998 nodes and 39,988 elements; and 1.0 rcl for 16,800 nodes and 11,154 elements.

To evaluate the convergence behavior of the macroscale stress and strain results regardless of the element size, the tension tests were simulated by applying a strain loading at the Z-direction at a strain of 0.2. The stress–strain curve for the tension simulation under various element size is provided in Figure 8, where the stress response in the elastic region at different element size follows the same path, by contrast, above the yield strength, there is various stress–strain simulation behavior affected by the different element sizes, such as 0.3 and 1.0 rcl have similar results, while 0.4 and 0.6 rcl are similar. Moreover, the stress difference and relative error between 0.4 and 0.6 rcl are less than 35 MPa and 4.2%, and the stress difference and relative error between 0.3 and 1.0 rcl are less than 15 MPa and 1.6%. Comparing these simulation results with experimental data presented in Figure 9, and more detailed data can be found in Table 6, and then 0.6 rcl is better than others and chosen in the current work.

### 4.2. Macromechanical Response and Strength Contribution

The evolution of heterogeneous macromechanical response for the MarBN steel computed by the macro-model is presented in Figure 9a. A good consistency (yield and ultimate tensile strength) between experimental and computed results can be observed, indicating that the effect of microstructure on the macromechanical properties can be considered in the macro constitutive model. Moreover, to clarify the mechanistic contributions to the strength of MarBN steel, a comparison diagram was presented, as shown in Figure 9b. Whereas different obstacle strengthening in MarBN steel presents various mechanistic representations. According to Figure 7b, the strength contribution of the solid solution reaches 400 MPa. The strength contribution of grain boundaries is about 140 MPa and dislocation density is about 80 MPa, while the strength contribution of Peierls–Nabarro is only about 40 MPa and that of the precipitates about 30 MPa. The strength contribution of interstitial atoms is about 20 MPa. Therefore, it is found that solid strengthening, grain boundaries, and dislocation density played a more crucial role than others. The mechanistic contribution caused by the various obstacle strengthening in MarBN steel is critical and should be addressed for material under service conditions with microstructure evolution, such as poor yield strength under high temperature resulting from the depletion of solute atoms and loss of dislocation pattern and density.

### 4.3. Micromechanical Response and Failure Mechanism

Figure 8 presents the local elastic strain along the loading direction under 1% (a) and 10% (b) total strain. The elastic strain under two various total strains is inhomogeneous, as shown in Figure 10a, where many grains present higher elastic strain than others, such as grains 1–15 marked by the red color. When the loading up to the 10% total strain, as shown in Figure 10b, the elastic strain of grains 1–15 is not only higher but also joined together, forming a high-strain region. The results indicate that the elastic strain is dependent on the grain orientation, resulting in inhomogeneous deformation.

Figure 11 shows the equivalent plastic strain along the loading direction under 1% (a) and 10% (b) total strain. Similar to the result of elastic strain, the equivalent plastic strain is also inhomogeneous, as shown in Figure 11a, where the equivalent plastic strain of grains 1–15 is also higher than others. In addition, from Figure 11b, plastic strain within the whole RVE model can almost be addressed. Interestingly, under 10% total strain, higher equivalent strain forms the strain bands with around 45 degrees marked by the red lines named 1 and 2, resulting in tensile shear failure. The result is in agreement with the tensile fracture characterization observed by SEM [4].

Figure 12 shows the equivalent strain along the loading direction under 1% (a) and 10% (b) total strain. A higher equivalent strain occurs at the grain boundaries, such as in grains 1–15. Under 10% total strain, the trend of higher equivalent strain is like the equivalent plastic strain, where the strain bands marked by the red lines named 1 and 2 are formed. This indicates that the strain bands are critical for crack initiation and tensile failure.

### 4.4. Failure Mechanism Evaluation and Microstructure Characteristics

To further reveal microstructure evolution and failure mechanism during tensile loading, many specific grains can be employed for the microstructure analysis using MTEX toolbox [63] in MATLAB based on EBSD results after tensile failure, as shown in Figure 13. From Figure 13a, the Schmid factor with slip system can be calculated, where the higher Schmid factor is located at the grain boundaries. The results agree with the strain distribution, as shown in Figure 12. Besides, the various slip systems are active in the different grains, resulting from the various grain orientations with inhomogeneous deformation during tensile loading. Figure 13b,c show the KAM and GNDs distribution, respectively. The higher value appears close to the grain boundaries and declines into the grain, resulting from the plastic strain along the grain boundaries presented in Figure 11.

Coupling simulation and microstructure by EBSD, the microstructure evolution can be well explained by the micromechanical model. Strain is related to the grain orientation, leading to inhomogeneous deformation, and then to the forming various Schmid factor and slip systems. Besides, plastic strain can happen close to the grain boundaries and decline into the grain, resulting in higher KAM and GNDs in the grain boundaries. The higher the loading, the higher the local strain. Then, the shear bands with around 45 degrees can be formed, resulting in tensile shear failure.

## 5. Conclusions

In the paper, a multiscale model was constituted with a macro-model considered by the physically based yield strength and a crystal plasticity model with micromechanical properties and realistic grain orientation based on the RVE. The main conclusions that were obtained are as follows:(1)The macromechanical response can be simulated by the physically based yield strength model, where solid strengthening, grain boundaries, and dislocation density played a more crucial role than others.(2)The micromechanical response can be addressed by the crystal plasticity model considering realistic grain orientation based on the RVE model, where the elastic strain is inhomogeneous due to the grain orientation, and many grains show higher elastic strain than others. Furthermore, the higher equivalent strain leads to the strain bands with around 45 degrees, resulting in tensile shear failure.(3)Regarding coupling simulation and microstructure by EBSD, the microstructure evolution can be well explained by the micromechanical model. As well, plastic strain can occur close to the grain boundaries and decline into the grain, resulting in higher KAM and GNDs in the grain boundaries proved by EBSD, resulting in crack initiation and shear failure.

## Figures and Tables

**Figure 1 materials-16-05194-f001:**
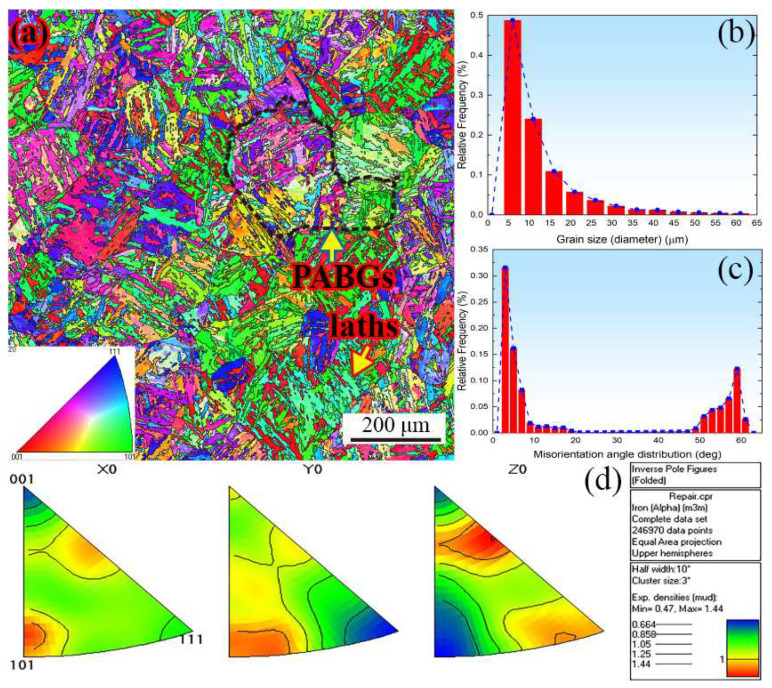
EBSD characterization of MarBN steel: (**a**) polo figure and grain boundaries; (**b**) grain size; (**c**) misorientation angle distribution; (**d**) inverse pole figure.

**Figure 2 materials-16-05194-f002:**
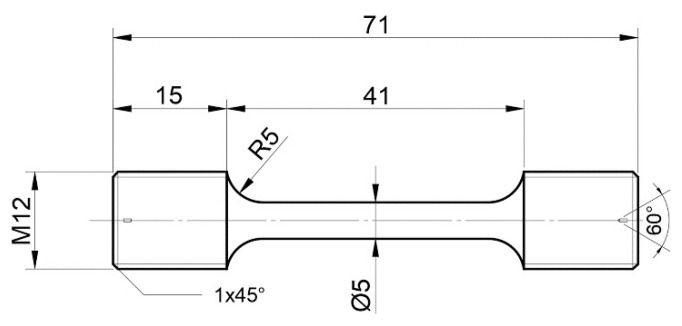
Sketch of uniaxial tension geometry (Unit: mm) [15].

**Figure 3 materials-16-05194-f003:**
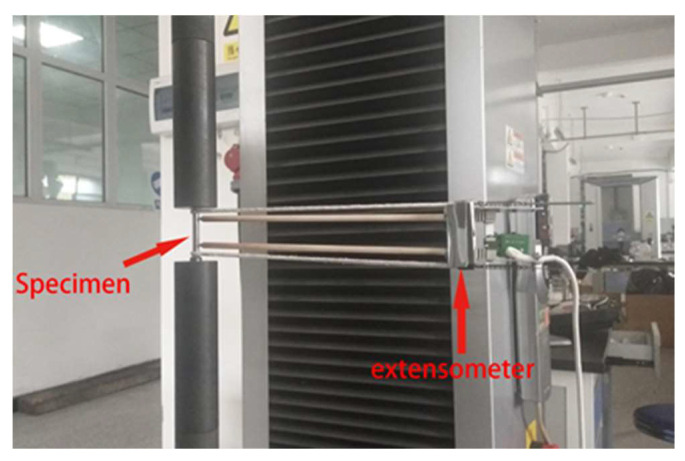
Tensile experimental device.

**Figure 4 materials-16-05194-f004:**
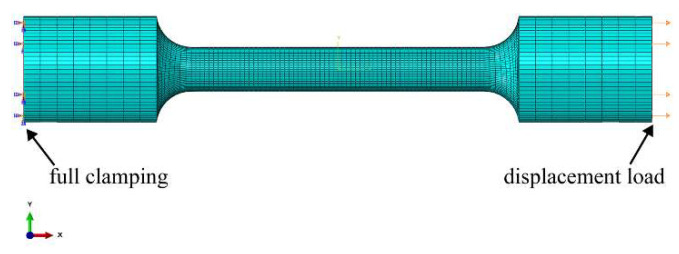
Macro finite element model of the tensile sample.

**Figure 5 materials-16-05194-f005:**
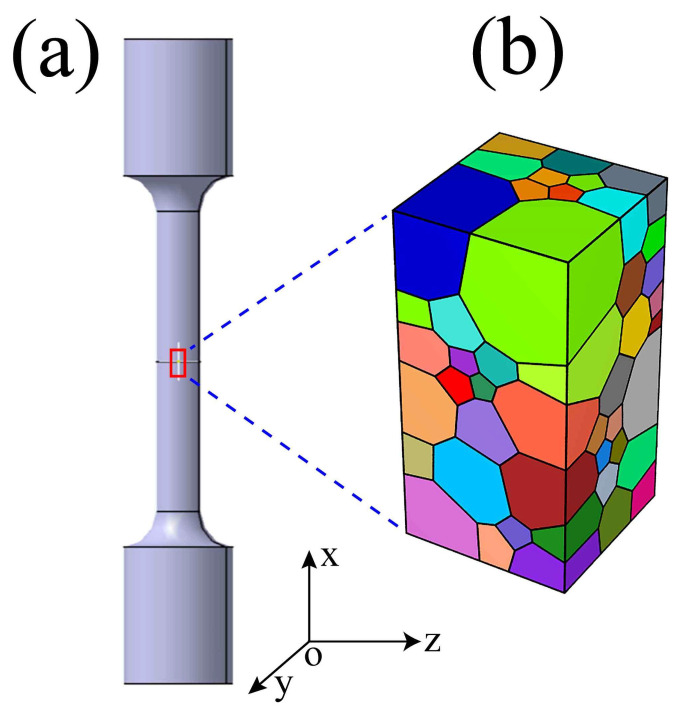
RVE model for crystal plastic finite element simulation: (**a**) macroscopic rod-like tensile specimen; (**b**) mesoscopic grain model.

**Figure 6 materials-16-05194-f006:**
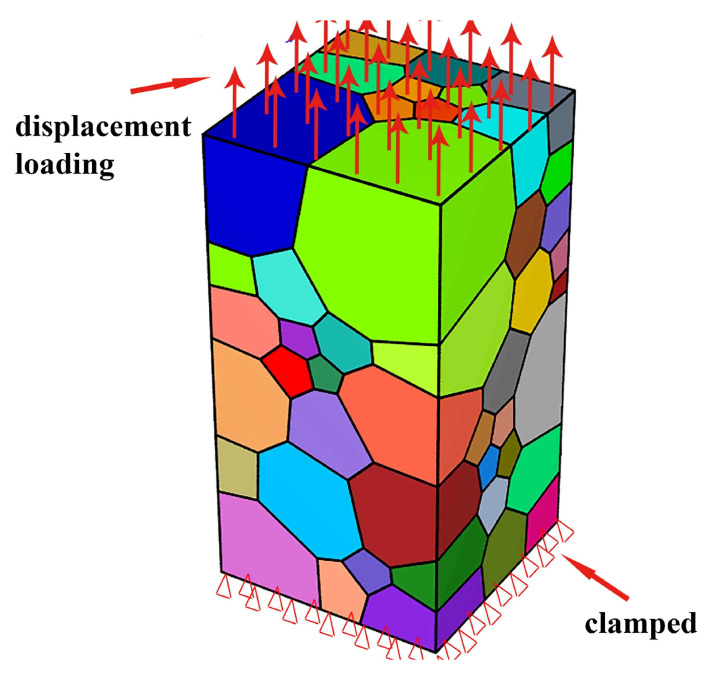
The boundary conditions of grain model.

**Figure 7 materials-16-05194-f007:**
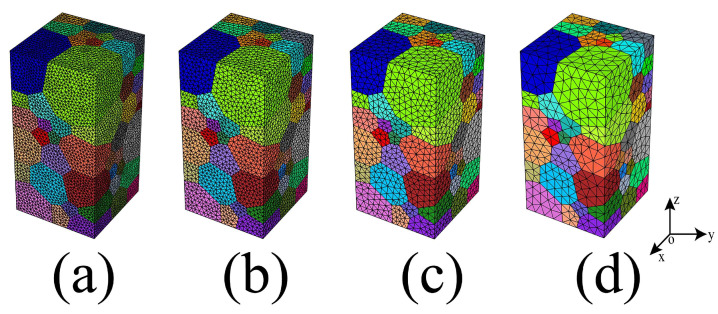
RVE model with various element sizes: (**a**) rcl 0.3 for 472,147 nodes; (**b**) rcl 0.4 for 197,660 nodes; (**c**) rcl 0.6 for 57,998 nodes; (**d**) rcl 1.0 for 16,569 nodes.

**Figure 8 materials-16-05194-f008:**
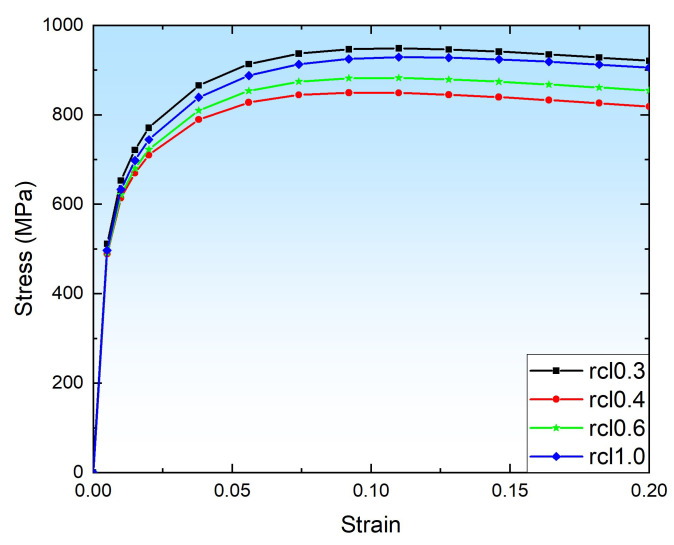
Stress–strain curve for the tension simulation with various element size.

**Figure 9 materials-16-05194-f009:**
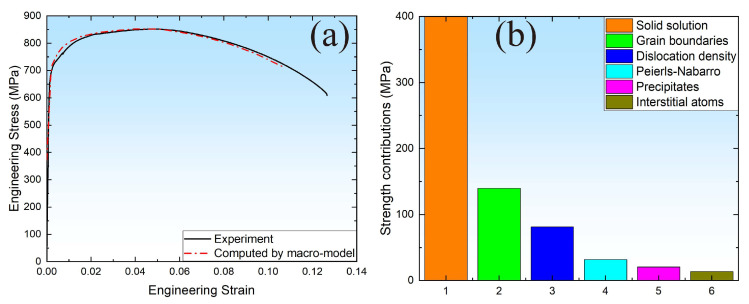
Comparison of the stress–strain curve with experimental tests and macro-model (**a**) and strength contribution (**b**).

**Figure 10 materials-16-05194-f010:**
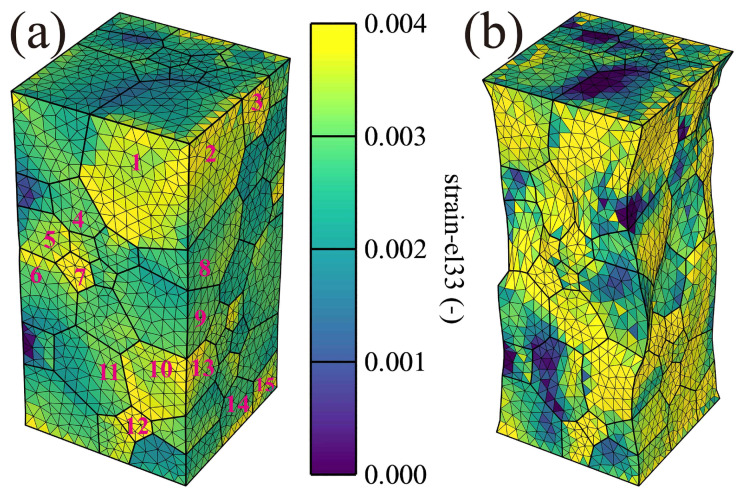
Elastic strain along loading direction (Z): (**a**) 1% and (**b**) 10% total strain for the real orientated grain distribution.

**Figure 11 materials-16-05194-f011:**
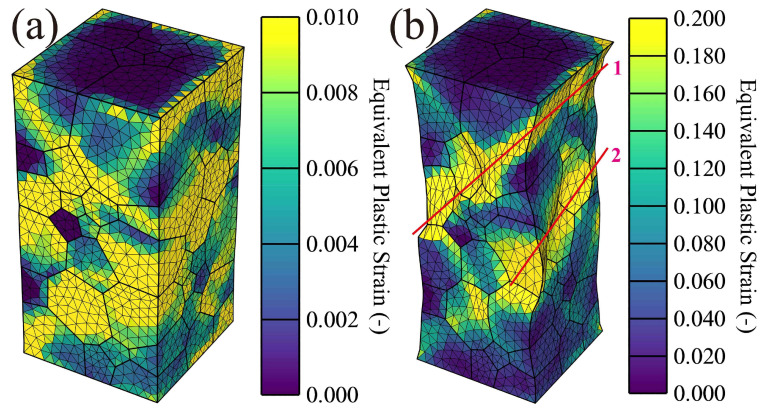
Equivalent plastic strain along loading direction (Z): (**a**) 1% and (**b**) 10% total strain for the real orientated grain distribution.

**Figure 12 materials-16-05194-f012:**
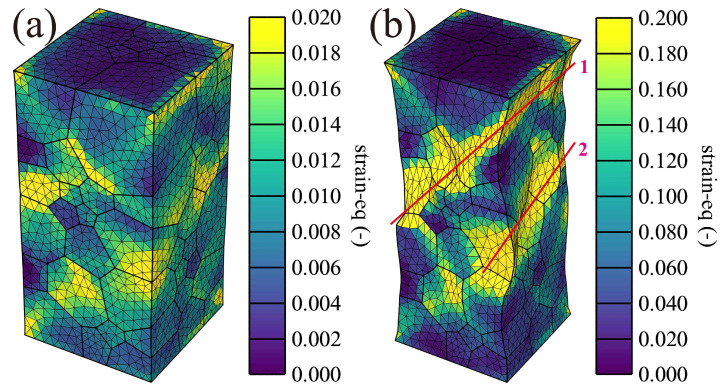
Equivalent strain along loading direction (Z): (**a**) 1% and (**b**) 10% total strain for the real orientated grain distribution.

**Figure 13 materials-16-05194-f013:**
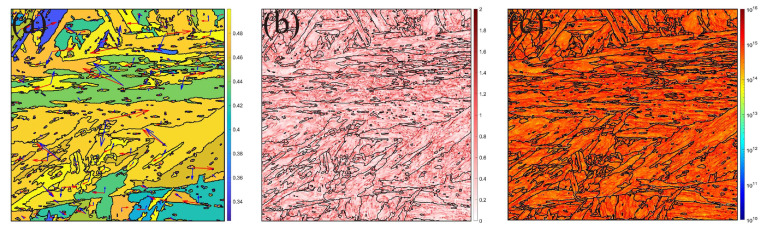
Microstructure by EBSD: (**a**) Schmid and slip index; (**b**) KAM; (**c**) GNDs.

**Table 1 materials-16-05194-t001:** The chemical composition of MarBN.

Composition	Cr	W	Co	Ni	V	Mo	Mn	C	Nb	Si	N	Fe
proportion (wt.%)	9.16	2.95	2.82	0.40	0.20	0.20	0.20	0.10	0.08	0.06	0.02	rest

**Table 4 materials-16-05194-t004:** Single crystal elastic constants.

Phase	Type	*C*_11_ (MPa)	*C*_12_ (MPa)	*C*_13_ (MPa)
M	BCC	$228.1\times 10^{3}$	$135.0\times 10^{3}$	$113.2\times 10^{3}$
α		$236.9\times 10^{3}$	$140.6\times 10^{3}$	$116.0\times 10^{3}$

**Table 5 materials-16-05194-t005:** Slip system strengths and plasticity parameters for various phases.

**Phase**	**m**	${\dot{\gamma }}_{0}$	$h_{0}$	$\tau_{0}$	$\tau_{s}$	n
*M*	0.05	0.001	40,000	680	700	1
*α*	0.05	0.001	4500	200	370	1

**Table 6 materials-16-05194-t006:** Comparing data under different rcl with experimental results.

Curve	Elastic Limit (MPa)	Maximum Stress (MPa)	Maximum Stress Difference (MPa)	Relative Error
rcl = 0.3	500.09	950.23	99.00	0.116
rcl = 0.4	498.78	810.34	40.89	0.048
rcl = 0.6	502.12	882.13	30.90	0.036
rcl = 1.0	501.34	927.34	76.11	0.089
Experimental curve	652.32	851.23		

## Data Availability

Data available on request from the authors.

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
