# Peer review of "Microstructure-Based Multiscale Modeling of Deformation in MarBN Steel under Uniaxial Tension: Experiments and Finite Element Simulations"

_materials, 2023, doi:10.3390/ma16145194_

Round 1
Reviewer 1 Report
This paper presents a study of the deformation of a MarBN steel based on experiments and the implementation of a multi-scale model to describe the behavior of the material. The aim of this work is not to develop behavior models or integrate them numerically into a calculation code, but to apply them to a particular material. The work is interesting but requires some improvements before it can be published.
1 – The introduction is quite short, but correctly presents the problem addressed and the difficulty of understanding the role of the microstructure in the deformation process of the material. Hence the importance of using numerical tools and a multi-scale behavioral model. At the macroscopic scale, the behavior is based on a physically based yield strength model, while at the microscopic scale, a crystalline plasticity model is used. At this small scale, the micro-mechanical properties and a realistic grain orientation of the RVE are considered.
2 – The second part concerns the experimental procedure with the study material and the tests.
On page 2, It would be clearer to present the composition of the alloy in a table. In addition, the white writing on the EBSD image in Figure 1 is very difficult to read.
3 – The third part is devoted to constitutive modeling. The models, although known from the literature, are complex and require many parameters. In the document, it is difficult to know which data are taken from the literature and which are measured/identified from experiments. This needs to be clarified, a summary table might help.
As an example, when presenting the macroscopic behavior model, the values of the coefficients E and µ seem to be taken from reference [13]. Is this true? Why are they given here? Were the same values measured on the grade studied?
4 – Results and discussion.
It is interesting to have the convergence study of the model. Convergence is examined by analysing the stress-strain curve simulated on the RVE model. The variable parameter is the relative cell volume rcl. It is clear that the curves obtained for different rcls are similar. However, it is difficult to know to what extent comparisons on the plastic part are made. Indeed, there is no clear trend in the result as a function of the rcl. Can you explain what criterion makes it possible to say that the rcl=0.6 value is the best choice?
In the images in Figures 7, 8 and 9, we can clearly see the influence of the RVE boundary condition (embedding of the upper and lower faces). Isn't this problematic for analyzing the deformation behavior of neighboring grains? Why not liberate the displacements in the x and y directions of the plane? Clamping a single point on the plane in the x and y directions is sufficient to ensure that the rigid body modes of the RVE are suppressed. What would be the influence of using a less intrusive boundary condition?
Some other comments:
· Acronyms must be defined the first time they appear in the abstract and in the text. To be done for KAM and GNDs in the abstract
· page 2, line 7, replace maro -> macro
· page 3, fig 1, the mark (a) on the EBSD image is not visible
· page 5, change the phrase: “Where A and b are a constant with a value of approximately unity with a value of 0.8 [16] and magnitude of Burgers vector of 0.286 nm [21], respectively…” -> “Where A is a constant with a value of approximately unity with a value of 0.8 [16] and b is the magnitude of the Burgers vector with a value of 0.286 nm [21].”
· page 13, fig 11, the marks (a, b, b) on the EBSD image are not visible
Reviewer 2 Report
The author needs to add more literature to the introduction section, and add the current year's literature also if available.
The literature gap and why the author required such a study had to be added to the manuscript.
Future scope needs to be added to the manuscript.
The Placement of Fig 1 and Fig 2 are different authors need to follow the journal guidelines.
The author needs to add the boundary condition Figure in the manuscript.
The author needs to check the formatting of the references as per the journal like 44,39,11,48,51 ….
An experiment Figure needs to be added
In Figure 7 author shows the comparison between the experiment and the computed results, accordingly, figure 7(b) needs to be explained. If the comparison of Figure 7 legend data like grain boundaries can be compared with the SEM micrograph of the experiment then it needs to be added to the manuscript.
Reviewer 3 Report
The authors conducted a study about experiments and simulations of MarBN steel under uniaxial tension. Microstructural investigations of deformation were made. The study is well organized and the subject of the manuscript is remarkable. However, in terms of evaluation and discussion, the manuscript seem week. Therefore, if the following regulations are made, the manuscript can be accepted for publication.
1. Demonstrate in the abstract novelty, practical significance. Abstract should be expanded sentences related to the results. The results of the study should be given as numerical percentages.
3. The introduction part is well-organized yet there is a reference problem. First, your reference list contains a few papers from electronics journal. If your work is convenient for this journal’s context then there are many references from this journal. Secondly, cited sources should be primary ones. Namely, indexed area shows the power of a paper and directly your paper’s reliability. Please make regulations in this direction.
4. It is necessary to give quantitative and qualitative indicators of the proposed method in conclusion. Conclusions should be written in more detail adding numeric data. Conclusions section is inadequate. There should be the importance of the study in detail, comparison results with other approaches in literature, the success of the prediction and computational results.
6. Improve the results and discussion and conclusion parts. Theoretical and experimental part of the study should be compared and evaluated detailly.
7. There are typographical and eventual language problems in paper.
8. It should be clearly stated the success rate of the study in abstract, results and discussion and conclusion parts.
The authors conducted a study about experiments and simulations of MarBN steel under uniaxial tension. Microstructural investigations of deformation were made. The study is well organized and the subject of the manuscript is remarkable. However, in terms of evaluation and discussion, the manuscript seem week. Therefore, if the following regulations are made, the manuscript can be accepted for publication.
1. Demonstrate in the abstract novelty, practical significance. Abstract should be expanded sentences related to the results. The results of the study should be given as numerical percentages.
3. The introduction part is well-organized yet there is a reference problem. First, your reference list contains a few papers from electronics journal. If your work is convenient for this journal’s context then there are many references from this journal. Secondly, cited sources should be primary ones. Namely, indexed area shows the power of a paper and directly your paper’s reliability. Please make regulations in this direction.
4. It is necessary to give quantitative and qualitative indicators of the proposed method in conclusion. Conclusions should be written in more detail adding numeric data. Conclusions section is inadequate. There should be the importance of the study in detail, comparison results with other approaches in literature, the success of the prediction and computational results.
6. Improve the results and discussion and conclusion parts. Theoretical and experimental part of the study should be compared and evaluated detailly.
7. There are typographical and eventual language problems in paper.
8. It should be clearly stated the success rate of the study in abstract, results and discussion and conclusion parts.
Round 2
Reviewer 1 Report
I approve the changes made and am in favour of publication as it stands.
Reviewer 2 Report
All the revisions are fine.